# Molecular engineering of organic semiconductors enables noble metal-comparable SERS enhancement and sensitivity

Gokhan Demirel [1]*, Rebecca L.M. Gieseking [2,3], Resul Ozdemir [1,4], Simon Kahmann [5], Maria A. Loi[5], George C. Schatz[2]*, Antonio Facchetti [2,6]* & Hakan Usta[4]*

Nanostructured molecular semiconductor films are promising Surface-Enhanced Raman Spectroscopy (SERS) platforms for both fundamental and technological research. Here, we report that a nanostructured film of the small molecule **DFP-4T**, consisting of a fully π-conjugated diperfluorophenyl-substituted quaterthiophene structure, demonstrates a very large Raman enhancement factor ($>10^5$) and a low limit of detection ($10^{-9}$ M) for the methylene blue probe molecule. This data is comparable to those reported for the best inorganic semiconductor- and even intrinsic plasmonic metal-based SERS platforms. Photoluminescence spectroscopy and computational analysis suggest that both charge-transfer energy and effective molecular interactions, leading to a small but non-zero oscillator strength in the charge-transfer state between the organic semiconductor film and the analyte molecule, are required to achieve large SERS enhancement factors and high molecular sensitivities in these systems. Our results provide not only a considerable experimental advancement in organic SERS figure-of-merits but also a guidance for the molecular design of more sensitive SERS systems.

[1] Bio-inspired Materials Research Laboratory (BIMREL), Department of Chemistry, Gazi University, 06500 Ankara, Turkey. [2] Department of Chemistry and the Materials Research Center, Northwestern University, 2145 Sheridan Road, Evanston, IL 60208-3113, USA. [3] Department of Chemistry, Brandeis University, 415 South Street, Waltham, MA 02453, USA. [4] Department of Materials Science and Nanotechnology Engineering, Abdullah Gül University, 38080 Kayseri, Turkey. [5] Zernike Institute for Advanced Materials, University of Groningen, Nijenborgh 4, 9747 AG Groningen, The Netherlands. [6] Flexterra Inc., 8025 Lamon Avenue, Skokie, IL 60077, USA. *email: nanobiotechnology@gmail.com; g-schatz@northwestern.edu; a-facchetti@northwestern.edu; hakan.usta@agu.edu.tr

Organic small molecules comprising π-conjugated moieties and functional groups have been the subject of considerable scientific and industrial research for applications ranging from pharmaceutical products and plasticizers to optoelectronics and sensors[1–5] as well as components of biological systems and natural products[6]. Despite all premises, π-conjugated small molecules have yet to play a significant role as substrate for surface enhanced Raman spectroscopy (SERS) studies. There are numerous reasons as why organic semiconductors might be considered as very promising building blocks to design next-generation Raman enhancement platforms. The presence of energetically accessible and delocalized π-electrons in these systems enables unique chemical, physical, and optoelectronic properties as compared with molecules having only σ-electrons[7]. Particularly, solid-state films of properly designed π-conjugated molecules are responsive to electromagnetic radiation in the visible/near-IR spectral region and can conduct electrical current under an external electric field, which brings very exciting opportunities for commercialization[8,9]. Since the solid state structure of π-conjugated molecules relies on relatively weak intermolecular interactions (e.g., Van der Waals, π–π, and C–H⋯π), facile film fabrication to form various micro/nanostructures is feasible by using simple solution- or physical vapor deposition (PVD)-based techniques[10,11]. π-conjugated systems could be biocompatible and they could have very low materials production costs once their synthesis processes are optimized. Highly delocalized molecular orbitals of π-conjugated systems ensure efficient orbital overlap with the analyte molecules, which could facilitate intermolecular charge transfer processes with the analyte molecules to contribute to the chemical enhancement mechanism in SERS. Very different than universal coinage metal surfaces, organic semiconductor structures could be specifically designed to detect a particular analyte molecule, which could lead to molecule-specific SERS platforms[12]. The possibility to tune organic semiconductor carrier density by molecular and field-effect doping processes could open new possibilities for sensor device architectures. In order to achieve large SERS enhancements, molecular structure of the organic semiconductors should be carefully designed and must provide finely tuned π-molecular orbital properties, electronic structures, charge transfer properties, and solid-state packing motifs, which are expected to affect the SERS activity of the organic semiconductor. Considering these striking advantages of molecular semiconductors, very recently we demonstrated[13] that metal-free nanostructured semiconductor films based on a π-conjugated small molecule, (**DFH-4T**, Fig. 1a), could be SERS-active with a remarkable enhancement factor (EF) surpassing $10^3$. Prompted by this breakthrough, we now question whether molecular engineering of the SERS-active organic semiconductors, and thus the corresponding optoelectronic properties and solid-state microstructure/morphology, can be tuned to further advance Raman enhancement performance.

Herein, we explore a fluoroarene-modified oligothiophene, 5,5‴-diperfluorophenyl-2,2′:5′,2″:5″,2‴-quaterthiophene (**DFP-4T**, Fig. 1a) semiconductor[14,15], which includes an electron-rich quaterthiophene π-core similar to that of **DFH-4T** but end-capped with π-electron-deficient perfluorophenyl (–C₆F₅) units. Thus, this molecule employs a quite different design approach as compared with **DFH-4T**, wherein the σ-insulating perfluoroalkyl substituents present in **DFH-4T** are removed and replaced with –C₆F₅ substituents leads to a fully π-conjugated backbone featuring small intramolecular torsions (*vide infra*). Furthermore, this molecular design enhances solid-state molecular packing and correspondingly favors very efficient charge-transfer/transport in films and facilitate frontier orbital wave function overlap with the analyte molecules to further enhance SERS performance. We preserved fluorine substituents on the outer benzene rings to

continue to provide excellent volatility for reliable and quantitative PVD film fabrication. Hydrophobic and nanostructured **DFP-4T** films prepared via PVD are found to exhibit very large EF values of $>10^5$, indicating that the SERS performance of metal-free organic semiconductor films now approach those of the current plasmonic metal and inorganic semiconducting platforms. Furthermore, a very low detection limit of $10^{-9}$ M is demonstrated for methylene blue (MB) as the probe molecule. Computational analysis demonstrates that the large EFs can be attributed to a resonance enhancement mechanism related to a weakly absorbing charge-transfer excited state that is near-resonance with the Raman laser excitation.

## Results

**Synthesis and fabrication of nanostructured DFP-4T films.** **DFP-4T** was synthesized by Pd-catalyzed Stille coupling and purified by gradient-temperature vacuum (~$10^{-6}$ Torr) sublimation according to the literature[16]. Nanostructured films of **DFP-4T** with a thickness of $1.1 \pm 0.2$ μm were then fabricated via PVD on Si(001) substrates under vacuum (~$10^{-6}$ torr) (Fig. 1a). The film deposition parameters were modified, versus those used for thin-film transistor fabrication, as short source-substrate distance of ~5–7 cm, low substrate temperature of ~30–40 °C, and ultrafast deposition rate of >40 nm s$^{-1}$ to promote **DFP-4T** out-of-plane crystal growth[17,18]. **DFP-4T** film morphology was characterized using field emission scanning electron microscopy (FE-SEM) and atomic force microscopy. FE-SEM (Fig. 1b) analysis revealed uniformly distributed and highly interconnected vertically aligned nanoplates with lateral sizes of $64 \pm 8$ nm. The wettability of **DFP-4T** films was accessed by water contact-angle measurements (Fig. 1c), indicating a hydrophobic surface with a contact angle of $128° \pm 3.7°$. This hydrophobicity is attributed to the combinations of perfluoroarene terminal units attached to a quarterthiophene π-core and the three-dimensional nanostructured surface morphology (root-mean-square (RMS) roughness ~51.2 nm, Supplementary Fig. 1a).

To gain insight into the film microstructural features, the deposited **DFP-4T** film was characterized by X-ray diffraction (XRD) measurements. As shown in Fig. 1d, a strong diffraction peak at $2\theta = 21.3°$ is observed corresponding to a very short periodicity (d-spacing) of 4.17 Å. This film phase does not correlate with any simulated diffractions of the **DFP-4T** single-crystal[16], and it is much smaller than the **DFP-4T** long-axis molecular length (24.2 Å, Supplementary Fig. 2). However, considering that it is similar to the short axis molecular length (5.4 Å, Supplementary Fig. 2) and typical π-π stacking distances (~3.5–4 Å), it is reasonable to conclude that **DFP-4T** molecules adopt an arrangement in the nanostructured film with their long axes on the substrate plane allowing π–π interactions in the out-of-plane direction (Fig. 1e). Although **DFP-4T** crystalline domains in the film and in the single-crystal differ, BFDH (Bravais–Friedel–Donnay–Harker) theoretical crystal morphology reveals that **DFP-4T** molecules exhibit a behavior to grow 2D crystalline nanoplates perpendicular to the long molecular axis (Supplementary Fig. 3). This further confirms that the proposed molecular orientation with the long molecular axes in the substrate plane is consistent with the observed vertically oriented nanoplate-based morphology. The analysis of the previously reported single-crystal structure of **DFP-4T** indicates the presence of intermolecular cofacial π-π interactions (~3.7–4.0 Å) between thienyl–thienyl and perfluorophenyl–thienyl units, which could facilitate efficient frontier orbital wave function overlaps with top-lying analyte molecules in the out-of-plane direction. Furthermore, note that **DFP-4T** molecular structure indicates that the thiophene-thiophene (0°/17.6° for inner/outer linkages) and

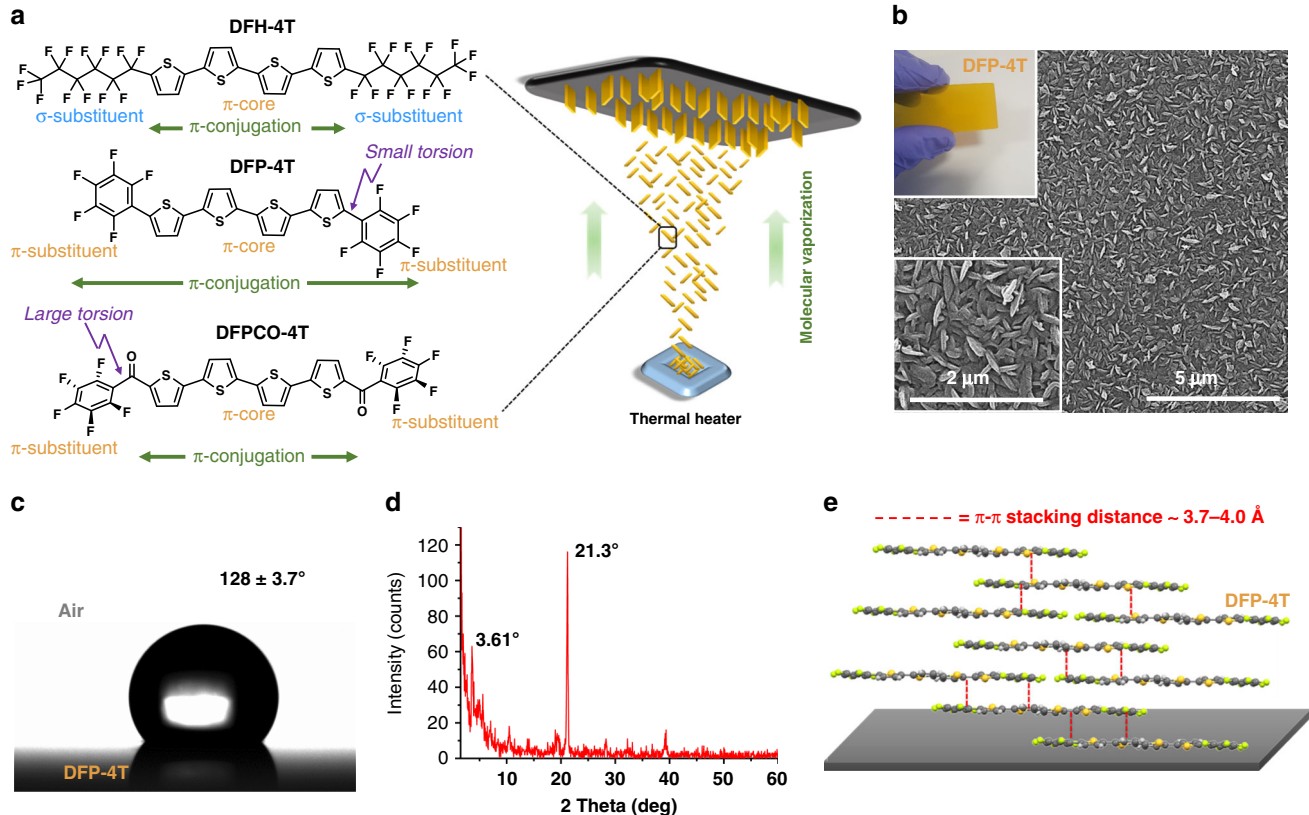

**Fig. 1** Fabrication and characterization of nanostructured **DFP-4T** films. **a** Chemical structures of the 4T-based semiconductors of this study depicting the extension of π-conjugation as a function of (σ- vs π-) substituents and torsion angles, and schematic illustration of the PVD process for fabricating the nanostructured SERS films. **b** SEM images of the **DFP-4T** films (top inset shows the optical image of the **DFP-4T** film). **c** Optical image of a water droplet on the **DFP-4T** film indicating the water contact-angle. **d** The θ−2θ XRD pattern of the **DFP-4T** film. **e** Schematic illustration of the proposed molecular packing diagram in the out of plane direction showing representative π-π interactions (red dashed lines).

thiophene-perfluoroarene (12.9°) torsional angles in **DFP-4T** are much smaller than the limiting ~30–40° value required for efficient intramolecular π-overlap to support a significant valence/conduction band structure[19,20]. A substantially planar **DFP-4T** molecular backbone on the crystallite surface also ensures efficient **DFP-4T**···**DFP-4T** intermolecular π-π stacking toward the crystal interior and **DFP-4T**···analyte molecule π–π stacking, which could amplify the molecular polarizability tensor of the analyte molecules in SERS measurements via photon-induced interfacial charge-transfer processes at the semiconductor surface. Finally, a far weaker low-angle secondary diffraction peak at 3.61° (d-spacing = 2.45 nm) corresponds to the **DFP-4T** molecular length (2.42 nm, Supplementary Fig. 2), indicating that a small portion of crystallites have the molecules standing upright along the substrate normal.

**Raman enhancement for nanostructured DFP-4T films.** The Raman enhancement ability of the **DFP-4T** films was investigated using the common probe molecule MB. MB is preferred as the Raman probe based on its known characteristic Raman bands and, more importantly, the fact that it does not exhibit an optical transition at the Raman laser excitation energy (1.58 eV) employed in this work neither in solution nor in the solid state (onset absorption energy for MB = 1.78–1.65 eV, Supplementary Fig. 4). Therefore, the contribution of the probe molecule's resonant excitation to the Raman enhancement can be excluded. While no SERS spectrum was recorded for a $10^{-3}$ M aqueous solution of MB on bare Si(001) substrate (Supplementary Fig. 5), substantial Raman signals were collected on nanostructured DFP-

4T/Si(001) substrates demonstrating that **DFP-4T** is SERS-active. Figure 2a shows the nonresonance Raman spectrum of MB measured on the nanostructured **DFP-4T/Si(001)** platform, which reveals good signal-to-noise ratios and peak positions consistent with earlier reports[13,18,21].

Note that the negative Raman intensities in spectra are due to the baseline correction. Several prominent bands originating from MB molecules are detected at 450 cm$^{-1}$ for (C–N–C) skeletal stretching, 1186 cm$^{-1}$ for (C–N) stretching, 1459 cm$^{-1}$ for (C–N) asymmetric stretching, and 1621 cm$^{-1}$ for (C–C) ring stretching. The significant Raman intensity enhancement for the peak at 1621 cm$^{-1}$ implies that the MB molecules adopt very favorable molecular orientations on the nanostructured **DFP-4T** film to promote substrate–molecule electronic interactions and charge-transfer processes[22,23]. To determine whether the nanostructured **DFP-4T** film contributes to these Raman peaks, a control experiment was carried out by collecting the Raman spectrum of pristine **DFP-4T** without any analyte (Fig. 2b). This data clearly indicate that the obtained Raman signals shown in Fig. 2a are exclusively based on the analyte MB molecules. The **DFP-4T** EF was calculated from the intensity magnification of the Raman peak at 1621 cm$^{-1}$ ($10^{-5}$ M MB concentration) taking the bare SiO$_2$/Si substrate as the reference (See "Methods" for details). Impressively, the resulting EF is $2.7 \pm 1.4 \times 10^5$, which is about 100 times higher than that we recently reported for **DFH-4T/Si (001)**[13] and ~10,000× larger than for other organic semiconductors[24]. The SERS performance of the current nanostructured **DFP-4T/Si(001)** platform is even comparable to those obtained with plasmonic structured metal films and the best inorganic semiconductor based SERS substrates[25–29]. The molecular

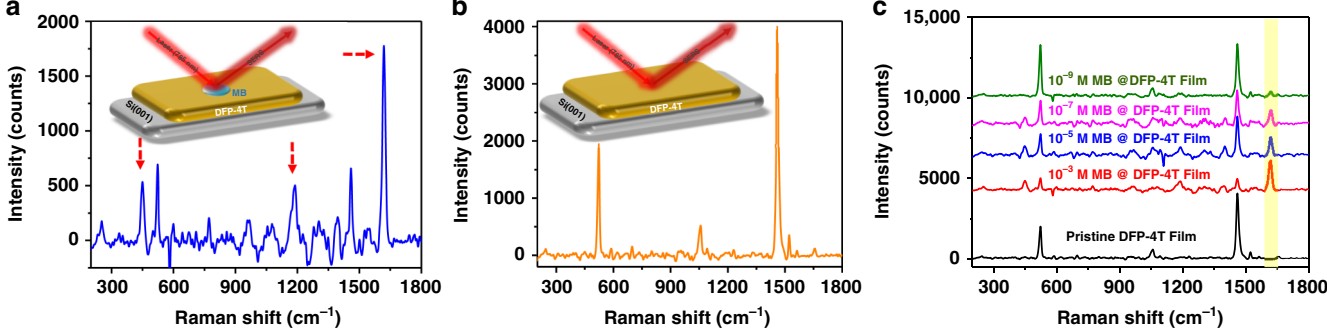

**Fig. 2** Raman enhancement for nanostructured **DFP-4T** films. **a** The SERS spectrum of MB probe ($10^{-3}$ M) on nanostructured **DFP**-4T film (characteristic Raman peaks are indicated). **b** Raman spectrum of pristine nanostructured **DFP-4T** film. **c** SERS spectra of MB on nanostructured **DFP-4T** film at different concentrations.

detection ability of the **DFP-4T** film was further studied by collecting SERS spectra at different MB probe concentrations from $10^{-3}$ to $10^{-9}$ M (Fig. 2c). An obvious Raman signal enhancement was observed even at a very low probe concentration of $10^{-9}$ M, indicating high detection sensitivity for the new organic semiconductor platform. Note that this is among the best limit of detection (LOD) values reported for a non-plasmonic SERS substrate[28–31].

**Effect of deconjugating DFP-4T on the SERS response.** To evaluate the role of the **DFP-4T** fully conjugated molecular structure on the observed high Raman enhancement performance, films of a structurally related organic semiconductor 5,5‴-di(per-fluorophenylcarbonyl)-2,2′:5′,2″:5″,2‴-quaterthiophene (**DFPCO-4T**, see structure in Fig. 1a), which includes carbonyl groups (C=O) between the fluoroarene units and the quaterthiophene core, were also investigated with SERS. The incorporation of electron-withdrawing carbonyl functionalities not only lowers the frontier molecular orbital (HOMO/LUMO) energy levels and changes their wavefunction topologies but also introduces local dipoles to manipulate molecular packing. The **DFPCO-4T** films, fabricated using the same PVD parameters as for **DFP-4T**, consist of mostly vertically aligned, interconnected nanoplates as evidenced by FE-SEM images (Fig. 3a). However, quite differently from the **DFP-4T** films, a notably larger density of in-plane grown domains are observed. The presence of both vertically aligned and flat grains together resulted in considerably higher surface roughness (RMS ~ 163 nm, Supplementary Fig. 1b) and, thus, enhanced hydrophobicity (water contact angle ~ 157° ± 2.4°, Fig. 3a). Going from **DFP-4T** to **DFPCO-4T**, the increased hydrophobicity could not simply be explained by molecular functionalization since adding polar carbonyl groups should reduce hydrophobicity. By analyzing the major diffraction peak observed at $2\theta = 23.7°$ (Fig. 3b) and using single-crystal structural parameters[32], **DFPCO-4T** dominant microstructure evolves along the [0 2 0] direction with the assistance of strong C–H···π contacts (~2.85–3.17 Å) in a herringbone fashion (Fig. 3d). The secondary peak observed at $2\theta = 18.2°$ corresponds to a periodicity along the [3 1 −1] direction that also includes the long molecular axis on the substrate plane exhibiting strong C–H···π contacts (~3.18–3.34 Å) in the out-of-plane direction (Fig. 3d). The perfluorophenyl end-units in **DFPCO-4T** are found to adopt highly twisted (~61°) orientations with respect to the quaterthiophene π-core, thus they are electronically disconnected to the core. Note that both the molecular coplanarity and the packing arrangement in **DFPCO-4T** film are also quite different from those of **DFP-4T**, which showed only strong π-π interactions between highly coplanar molecules in the out of plane direction. The drastic changes in molecular structure and packing between the two semiconductors

are undoubtedly caused by the carbonyl functionalization. The packing motifs observed in the **DFPCO-4T** films promote strong intermolecular interactions and favorable crystal growth in the out of plane direction, consistent with the observed morphologies by FE-SEM. Nanoplate-like morphology for **DFPCO-4T** film is predicted by BDFH theoretical modeling, and it also indicates that the 2D crystal plane grows along the [0 2 0] and [3 1 −1] directions (Supplementary Fig. 6). The presence of a small shoulder at a low angle of $2\theta = 5.03°$ ($d$-spacing = 1.7 nm) is attributed to the in-plane grown crystalline domains. SERS measurements with MB ($10^{-3}$ M) show that **DFPCO-4T** films are also SERS-active. However, as compared with **DFP-4T** films, all Raman signal intensities for the probe molecule substantially decrease resulting in an EF of $6.1 \pm 2.3 \times 10^3$ (Fig. 3c), which is similar to that of **DFH-4T**[13]. Thus, although both molecules share the same perfluorophenyl termini and a quaterthiophene π-core, carbonyl functionalization is detrimental (~100× decreased EF) to the Raman enhancement performance, likely because of the large C=O induced torsions that deconjugate the core from the perfluoroarenes.

**Spectroscopic analysis and charge transfer state.** Considering that our organic semiconductor films have a low intrinsic carrier density[8] and negligible plasmon resonances in the conduction band[33,34], the significant Raman enhancements observed for **DFP-4T** films are very unlikely to originate from the electromagnetic mechanism. In addition, the optical absorptions of both analyte (MB) and organic semiconductor films at the laser excitation energy (1.58 eV) (Supplementary Fig. 4) do not exist, which excludes the possibility of having resonant charge excitation processes on pristine analyte or semiconductor molecules. However, hybridized analyte/semiconductor electronic states present at the interface may eventually contribute to the chemical enhancement mechanism by magnifying the Raman scattering cross-sections. Moreover, these resonances can couple with other resonances such as 3D morphology-driven Mie scattering resonances, nanostructured semiconductor exciton (band gap) resonances, and analyte molecular resonance[8,35]. The charge-transfer resonances enhance the polarizability derivative tensor for analyte molecule vibrational modes and consequently amplify the Raman signal intensity. These assumptions are consistent with the resonance description of SERS proposed by Lombardi and Birke[36].

To investigate the analyte-semiconductor charge-transfer processes, photoluminescence (PL) spectroscopy measurements were carried out on the current analyte-semiconductor platforms. The steady-state PL spectra of the pristine and MB-coated ($10^{-3}$ M) **DFP-4T** films were investigated under illumination at 400 nm, where substrate excitation is the strongest (Supplementary Fig. 4).

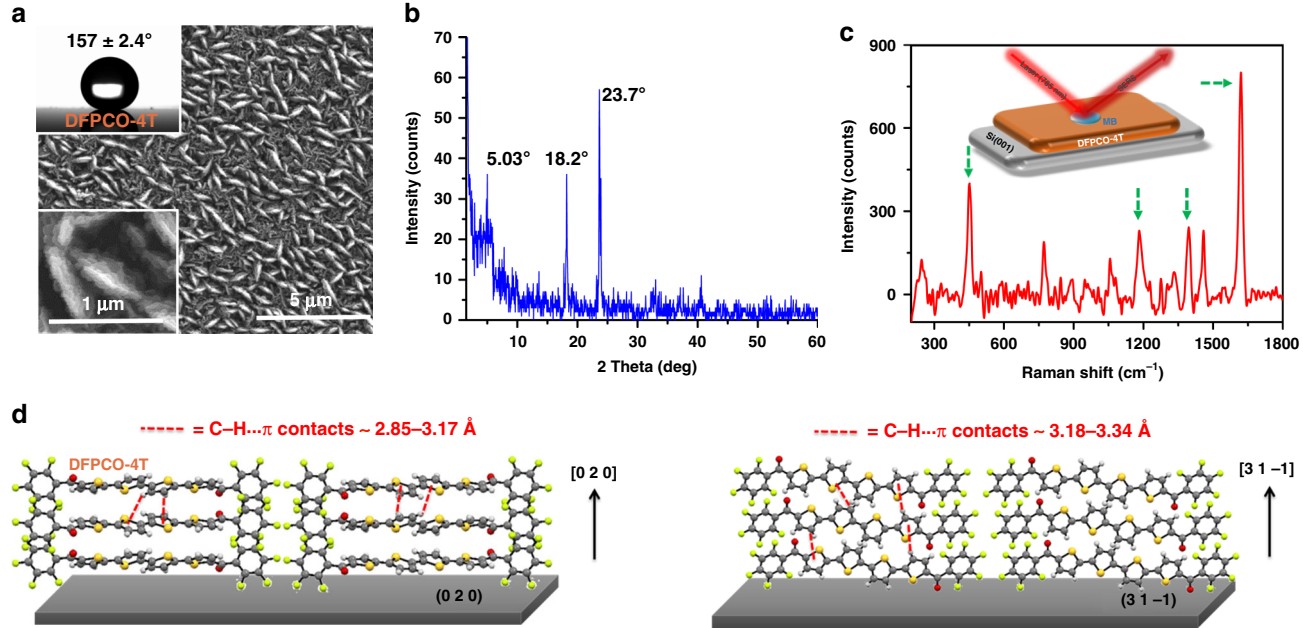

**Fig. 3** Characterization and SERS response of **DFPCO-4T** films. **a** SEM images of nanostructured **DFPCO-4T** films and, inset, optical image of a water droplet on **DFPCO-4T** film. **b** The $\theta$–$2\theta$ XRD pattern of **DFPCO-4T** film. **c** SERS spectrum of MB ($10^{-3}$ M) on **DFPCO-4T** (the characteristic peak positions are highlighted). **d** Schematic illustration of the molecular packing diagram in the out of plane direction ([0 2 0] and [3 1 −1]) simulated from the single-crystal structure parameters showing representative C−H···$\pi$ interactions (red dashed lines).

For the **DFP-4T**, the PL spectra for both samples are virtually identical with the emission entirely originating from **DFP-4T** (Fig. 4a). The transient PL spectra for both samples were measured in different positions and MB deposition was found to consistently reduce the PL lifetimes at short delays (Fig. 4b). This suggests that although emissive MB states are not populated, the presence of MB on **DFP-4T** provides additional decay pathway(s) for the photoexcitation, presumably via charge-transfer involving MB electronic states. Energy transfer can be excluded as there is no signature of the MB PL in the spectra. The **DFPCO-4T** system exhibits a pronounced intensity increase in the tail of the PL upon the presence of MB film. Similar to the case above, the main emission peak resembles the spectrum of the pristine substrate and a reduction of the PL lifetime can be observed in Fig. 4c (albeit weaker in this case). Again this suggests the presence of a charge-transfer towards MB. The region of increased intensity in the tail around 750 nm shows the inverse trend with a lifetime of around 1.38 ns for the MB/**DFPCO-4T** sample and 1.06 ns for the bare substrate. We thus attribute this observation to the presence of a charge-transfer state between MB and **DFPCO-4T**, as similarly observed before for organic-organic interfaces[37–39]. As we shall show below, the likely reason for the absence of such an effect in the MB/**DFP-4T** system is due to its lower position in energy.

**Raman enhancement mechanism and molecular design**. To gain insight into the Raman enhancement mechanism in these systems, we have computed the excited-state properties of complexes formed by the analyte (MB) and organic semiconductors (see Methods for details). Our computational results at the INDO/SCI level suggest that the SERS enhancements in **DFP-4T**/MB and **DFPCO-4T**/MB are due to the presence of charge-transfer (CT) excited states with a small but non-zero absorption cross-section near resonance with the Raman laser (Table 1). This result is consistent with our previous computational results for MB on **DFH-4T**[13]. In all three complexes, the CT excited states

have >95% charge-transfer character from the HOMO of the semiconductor molecule to the LUMO of MB. Interestingly, the CT excited state energies strongly correlate with the substrate HOMO energies. Note, as discussed earlier, the addition of carbonyl functionalities stabilizes the frontier molecular orbitals of **DFPCO-4T**, which causes the complex involving this semiconductor to have the highest CT excited state energy.

We now consider the SERS enhancements for MB on these three substrates on resonance with the CT energy of each substrate. As shown previously[13], a lifetime broadening of $\Gamma = 0.025$ eV is required to reproduce the EF of MB on **DFH-4T**; thus here we use this broadening to compute all the Raman intensities reported here. The predicted EF for **DFP-4T**/MB is a factor of six larger than that for **DFH-4T**/MB, whereas the predicted EF for **DFPCO-4T**/MB is slightly smaller than that for **DFH-4T**/MB (Table 1). For all three complexes, the CT state absorbs light very weakly; this weak absorption is sufficient to cause large resonance Raman enhancement because the large change in charge density upon excitation leads to large changes in resonant polarizability derivatives with respect to vibrational motion. The CT state in the **DFP-4T**/MB complex has an oscillator strength more than double that of the other two complexes, leading to a larger EF. In addition, even though the CT state absorbs weakly, its energy is highly sensitive to changes in the molecular geometry, which means that it contributes strongly to the resonance differential polarizability. Our results also suggest that substrate morphologies that facilitate significant $\pi$-stacking between analyte and substrate molecules is critical to obtain large Raman enhancements, as obtaining a large oscillator strength for the CT state requires good overlap of the substrate and analyte wavefunctions. The experimental observation that the 3D nanostructured substrate enables chemisorption of the analyte molecules with the correct orientations to facilitate CT suggests that our computations showing a critical role of the CT states for Raman enhancement are reasonable. Although the 3D nanostructured substrate morphology may also provide weak electromagnetic enhancements[40], electromagnetic terms cannot be the dominant

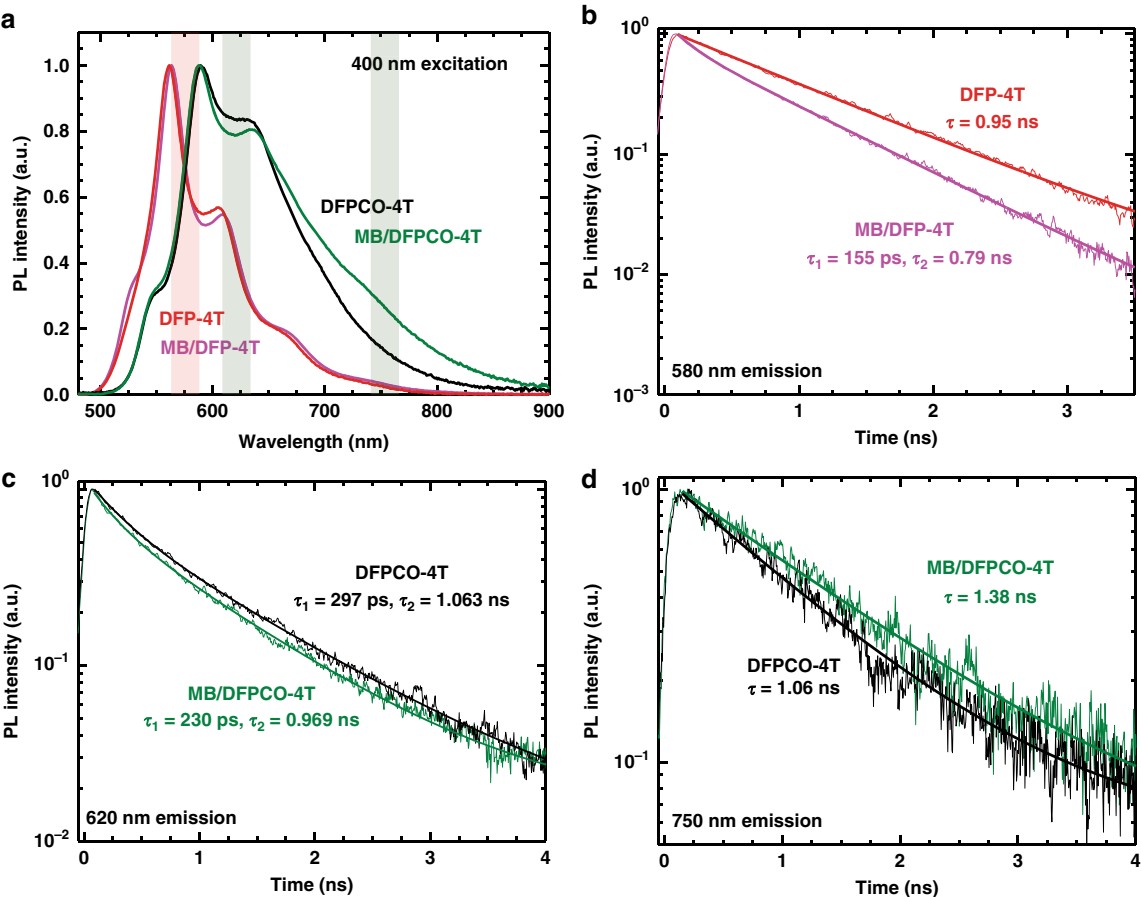

**Fig. 4** Photoluminescence spectroscopy measurements. **a** Steady-state and (**b**–**d**) transient photoluminescence of films upon 400 nm excitation (The shaded areas in **a** indicate the spectral regions wherein PL dynamics were investigated in **b**–**d**). The steady state spectra of the **DFP**-**4T** system are virtually identical in both cases (**a**), but a quicker decay of the main emission in the transients of MB/**DFP**-**4T** points to a charge transfer towards MB (**b**). The **DFPCO**-**4T** system shows a marked increase in the PL around 750 nm upon MB deposition. The main emission occurs slightly faster **c** for MB/**DFPCO**-**4T**, but significantly slower in the tail region **d**.

**Table 1 Computational data[a] for the semiconductors and the semiconductor/MB complexes.**

| Semiconductor | HOMO Energy (eV) | CT Excited State Energy (eV) | CT Excited State Oscillator Strength | Computed EF | Experimental EF |
|---|---|---|---|---|---|
| DFP-4T | −6.91 | 1.35 | 0.024 | $1.3 \times 10^4$ | $2.7 \pm 1.4 \times 10^5$ |
| DFPCO-4T | −7.24 | 1.62 | 0.009 | $6.2 \times 10^2$ | $6.1 \pm 2.3 \times 10^3$ |
| DFH-4T | −7.10 | 1.43 | 0.011 | $2.3 \times 10^3$ | $3.4 \pm 1.3 \times 10^3$ |

[a]Highest occupied molecular orbital (HOMO) energies of isolated semiconductor molecules, CT excited state energies, CT excited state oscillator strengths, and SERS enhancement factors on resonance with the CT excited state for the semiconductor/MB complexes at the INDO/SCI level

effect; if this were the case, the Raman signal of **DFP**-**4T** would be much stronger than that of MB, which is contrary to the experimental results in Fig. 2.

Due to the limitations of our computational model, these computations show the overall trend of a very large EF for MB on **DFP**-**4T** seen experimentally but only capture part of the two orders of magnitude difference in the EF values between **DFP**-**4T** and the other semiconductors seen experimentally. One key limitation is that this model only considers interactions of one semiconductor molecule with MB and thus it is unable to capture effects related to cooperative interactions between adjacent semiconductor molecules. It's noteworthy that because of the high computational cost of the frequency calculations, computations with more than one substrate molecule are not feasible. The lack of an experimental value for the excited state width, and how

it might vary for the different systems we studied is another point that will have to be addressed. In addition, we compute the electronic structure of a single interaction geometry of MB with each substrate molecule. In real systems, there is likely a distribution of interaction geometries corresponding to multiple EFs. Since large CT resonance enhancement requires good spatial overlap between the semiconductor molecule and MB frontier molecular orbitals, differences in the semiconductor effective π-conjugation lengths affect what distribution of interaction geometries can lead to large SERS enhancement. Since **DFP**-**4T** has a nearly coplanar π-framework with the HOMO extending across the entire molecule, including the perfluorophenyl end groups (Fig. 5a), this data suggest that MB may interacting strongly with any part of the **DFP**-**4T** molecule, leading to a large overall EF independently of the specific geometry used in the

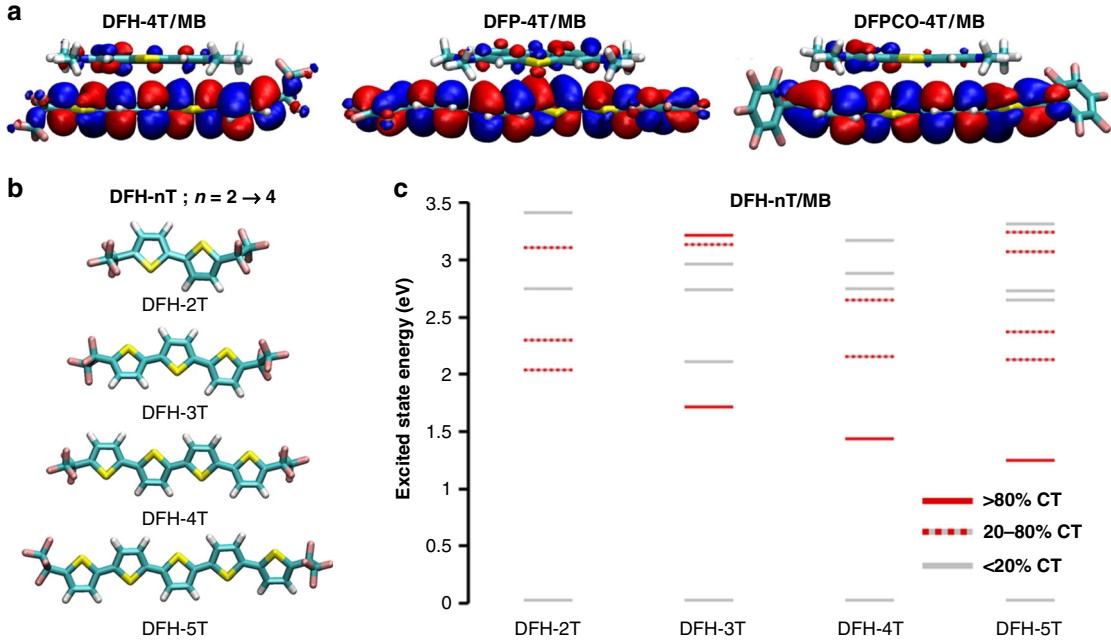

**Fig. 5** Computational analysis. **a** Highest occupied molecular orbital (HOMO) topologies of the indicated semiconductor/MB at the INDO/SCI level. **b** Optimized geometries at the ωB97X-D/cc-pVTZ level of **DFH-nT** molecules studied to assess the effect of the π-backbone length. **c** Excited-state energies of **DFH-nT**/MB complexes as a function of the nT extension, computed at the INDO/SCI level. The color of each state indicates its charge-transfer (CT) character.

computations. In contrast, for **DFPCO-4T**, steric repulsion between the terminal thiophene ring and the perfluorophenyl end group prevents planarization of the terminal groups and localizes the HOMO on the central quaterthiophene π-core, leading to a similar effective conjugation length of **DFH-4T**. Thus, a MB molecule interacting with the end groups of **DFPCO-4T** or **DFH-4T** will be unable to effectively couple with the entire molecular substrate reducing the EF. These geometric differences may account for some of differences in EFs not captured by our simple computational model. In addition, we compute all enhancement factors (EFs) on resonance with the CT states; if the CT states in the experimental systems somehow differ in how close to resonance they are with the Raman laser, this will likely cause some deviation in the trends in EFs. We note that our computational model reverses the ordering of the EFs on **DFPCO-4T** and **DFH-4T** relative to their experimental values, likely due to some combination of the semiconductor/**MB** molecular packing parameters and resonance conditions described here.

Finally, to provide a guideline for future molecular designs, we have also evaluated the effect of the semiconductor π-conjugated backbone molecular length on the SERS EF. We have modeled the **DFH-nT** (n = 2–5; Fig. 5b) semiconductors and their complexes with MB. It is found that as the molecular length and the effective conjugation length increase, the HOMO energy of the isolated semiconductor molecule increases from −7.86 eV to −6.80 eV (Supplementary Table 1). This change in the electronic structure has a dramatic effect on the excited-state properties of the semiconductor/MB complex. The energy of the CT excited state decreases from 2.04 eV to 1.25 eV, mirroring the magnitude of the change in the **DFH-nT** HOMO energies (Fig. 5c). We note that for **DFH-2T**, there is significant mixing of CT excitations with the local molecule excited states; the CT excited state we identify here has only 51% CT character. The oscillator strengths for the CT excited states are relatively small (Supplementary Table 1), with the exception of **DFH-2T**/MB; this is due to the aforementioned mixing of CT and local excitations.

Employing a Raman excitation source on resonance with the CT state, the SERS EF of the MB characteristic mode tends to increase with the **DFH-nT** conjugation length, with the exception of **DFH-2T**. However, this strong enhancement requires resonance of the incoming light with the CT state; the EF is quite small when the Raman excitation source is detuned from resonance with the CT state. These results suggest that two features are required to obtain large SERS EFs in organic systems: (1) the CT energy must be tuned such that it is very close to resonance with the incoming light, and (2) the semiconductor must be able to effectively π–π-stack with the probe molecules to obtain a small but nonzero oscillator strength in the CT state.

## Discussion

An efficient SERS platform has been developed by employing an organic semiconductor molecule, **DFP-4T**, through a simple vapor deposition technique. The resulting **DFP-4T** films demonstrate an outstanding SERS enhancement ability up to $2.7 \times 10^5$ and a LOD as low as $10^{-9}$ M, which can be comparable to the best inorganic semiconductor and even plasmonic metal SERS platforms. The high SERS enhancement of **DFP-4T** films appears to be governed by a resonance charge-transfer mechanism between the **DFP-4T** film and the probe MB molecule. This mechanism was further supported by the spectroscopy experiments and electronic structure calculations. These observations offer not only valuable information to the future design of organic semiconductor-based SERS platforms, but also provide an important advance in the applicability of SERS to the field of bio-/chemical sensors. As potential future directions to this end, chemical doping or electrical charge-injection (under gate-bias for example) could tailor band edges and introduce free charge carriers into organic semiconductor structures, which could lead to the formation of localized surface plasmons. This could potentially be useful to further advance the SERS EFs of organic semiconducting platforms. Note that it is known in the literature that localized surface plasmons could be generated in non-metallic structures such as inorganic semiconductors[34]. On the

other hand, since π-conjugated frameworks of organic semiconductors offer a great platform that allows for almost unlimited structural variations, their chemical structures could be specifically tailored to optimize the charge-transfer to a particular analyte molecule.

## Methods

**Fabrication/characterization of organic semiconductor films**. 5,5‴-Diperfluorophenyl-2,2′:5′,2″:5″,2‴:5‴,2‴″quaterthiophene (**DFP-4T**) and 5,5‴-bis(perfluorophenylcarbonyl)-2,2′:5′,2‴-quaterthiophene (**DFPCO-4T**) molecules, which were used as molecular building blocks in PVD, were synthesized and purified in accordance with previously reported procedures[16,32]. Prior to the nanostructured semiconductor film fabrication, silicon wafers (001 crystallographic orientation and 1–10 Ω resistivity) were cleaned with acetone and ethanol (10 min sonication) that was followed by a treatment with piranha solution (1 h). Finally, precleaned samples were further treated with UV–ozone for 15 min. **DFP-4T** or **DFPCO-4T** powder (10–20 mg) was placed onto a tungsten boat and thermally evaporated under a fixed base pressure of $1 \pm 0.2 \times 10^{-6}$ torr in a conventional PVD system (NANOVAK HV) using 90° deposition angle and ultrafast deposition rate (>40 nm s$^{-1}$). During the deposition process, the distance between the semiconductor source and the substrate was kept at ~5–7 cm. The morphologies and microstructures of the as-deposited films were characterized by a scanning electron microscope (FIE QUANTA 400 F Field Emission SEM) and XRD (Rigaku Ultima-IV X-ray diffractometer). UV–vis absorption and water contact angle measurements were performed using a Shimadzu 2600 UV–vis–near-IR spectrophotometer and a Krüss, DSA 100 drop shape analyzer, respectively. PL spectra were recorded at 400 nm excitation using the second harmonic of a mode-locked Ti:sapphire laser (Mira 900, coherent) at a repetition rate of 76 MHz. Steady-state spectra were recorded with a Hamamatsu EM-CCD camera. The excitation beam was spatially limited by an iris and focused with a 150 mm focal length lens. The fluence was adjusted using gray filters and spectra were taken in reflection geometry. Time-resolved traces were recorded with a Hamamatsu streak camera in single sweep mode. An optical pulse selector was used to vary the repetition rate of the exciting pulses.

**SERS experiments**. A Delta Nu Examiner Raman microscope with a 785 nm laser source was used for Raman investigations with an operational range of 200–2000 cm$^{-1}$. Unless otherwise specified, all Raman measurements were performed under the same experimental conditions (i.e., ×20 microscope objective with a 3 μm laser spot size, 150 mW laser power, and 30 s acquisition time). To evaluate the SERS enhancement performance of the organic semiconductor films, MB, a Raman probe, was drop-casted (5.0 μL, $10^{-3}$–$10^{-9}$ M in water) and kept in a hood until dry. The Raman spectra were then obtained from at least ten different locations of the dried spot area for each sample. The SERS enhancement of fabricated organic semiconductor films was evaluated by calculating the EFs using the equation (1). In calculations, the Raman spectra of MB (0.1 M) on the pristine silicon substrate was used as a reference.

$$\mathrm{EF} = (N_{\mathrm{Reference}} \times I_{\mathrm{Film}})/(N_{\mathrm{Film}} \times I_{\mathrm{Reference}}), \quad (1)$$

where the $N_{\mathrm{Reference}}$ and $N_{\mathrm{Film}}$ are the total number of MB molecules located in the laser spot area (7.065 μm$^2$) on the reference silicon substrate and on the nanostructured organic semiconductor films, respectively. The $I_{\mathrm{Reference}}$ and $I_{\mathrm{Film}}$ are the Raman peak intensities of 0.1 M MB on the reference silicon substrate and $10^{-5}$ M MB on the nanostructured organic semiconductor films at 1621 cm$^{-1}$, respectively.

The absolute Raman cross-section of MB on **DFP-4T** was also calculated. Rhodamin 6G (R6G) was selected as a standard to determine the cross-section of MB, and its Raman peak at 1510 cm$^{-1}$ was used as a reference with a non-SERS absolute cross-section of $9.6 \times 10^{-27}$ cm$^2$ sr$^{-1}$ at 785 nm[41] (extrapolated from 633 nm, assuming absolute cross-section is proportional to $\nu_S^3 \nu_L$)[42]. The absolute SERS cross-section of MB was then determined as $2.4 \times 10^{-24}$ cm$^2$ sr$^{-1}$ for the 1621 cm$^{-1}$ Raman peak.

**Computational methods**. The geometries and vibrational modes of the MB molecule, isolated substrate molecules, and complexes of MB with the substrate molecules were computed using density functional theory (DFT) with the ωB97X-D functional[43,44] and cc-pVTZ basis set[45], which were chosen for consistency with our previous computational results[13]. To address the effects of end groups, three substrate molecules were considered: **DFH-4T**, **DFP-4T**, and **DFPCO-4T**. To address the effect of substrate molecular length, a series of four substrate molecules with **DFH** end groups: **DFH-nT** ($n = 2$–5), where $n$ is the number of thiophene units in the molecular backbone (shown in Fig. 5b). We note that the **DFH-4T** substrate molecule is included in both series. To reduce computational costs, the perfluorohexyl substituents of **DFH-4T** were replaced with perfluoroethyl substituents. These calculations were performed with QChem 4.4[46]. The computed vibrational frequencies were scaled by the standard scaling factor of 0.956 for this method[47].

The excited-state energies were computed using the semiempirical Intermediate Neglect of Differential Overlap (INDO) Hamiltonian[48] with single-excitation

configuration interaction (SCI), consistent with our previous computations[13]. All possible single excitations within the minimal INDO basis set were generated, and the 7000 lowest-energy electron configurations were included in the CI matrix. The CI matrix was then diagonalized to yield the 2000 lowest-energy excited states. These calculations were performed using a home-built code based on portions of MOPAC 2016[49] and the INDO/CI code written by Jeffrey Reimers[50]. The INDO/SCI Raman intensities were computed using our DFT-based geometries, normal coordinates, and vibrational frequencies via numerical differentiation as have been previously described[13].

## Data availability

The data that support the findings of this study are available from the corresponding author upon reasonable request. The X-ray crystallographic coordinates for DFP-4T and DFPCO-4T have previously been deposited at the Cambridge Crystallographic Data Centre (CCDC) under the deposition numbers of 207396 and 288774, respectively. These data can be obtained free of charge from The Cambridge Crystallographic Data Centre via www.ccdc.cam.ac.uk/data_request/cif.

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

## Acknowledgements

This work was supported by Gazi University (Grant No: 05/2015-19), TUBITAK (Grant No: 117F468), and Flexterra Corporation. S.K. thanks the Deutsche For-schungsgemeinschaft for a research fellowship (Grant No: 408012143). R.L.M.G. and G.C.S. were also supported by NSF Grant CHE-1760537.

## Author contributions

G.D., H.U., and A.F. conceived and designed the experiments. H.U., R.O., and A.F. synthesized the small molecular organic semiconductors. G.D. fabricated the organic semiconductor platforms and performed the experiments. G.C.S. and R.L.M.G. designed and performed the theoretical calculations. S.K. and M.A.L. performed the PL experiments. G.D., H.U., A.F., M.A.L. and G.C.S. co-wrote the paper. All authors discuss the results and commented on the manuscript.

## Competing interests

The authors declare no competing interests.
