## [Peer Review File · Nature Communications]

Reviewers' comments:

Reviewer #1 (Remarks to the Author):

This paper demonstrates a notable Raman enhancement factor in films of DFP-4T, where the previously studied DFH-4T has been modified with electron-deficient groups. SERS spectra show the enhancement of the Raman signal. Photoluminescence and quantum-chemical calculations illustrate the role of CT states. Overall I find the data convincing, the significance high in that the EF approaches that of plasmonic systems, and the paper well written. Therefore I support publication in Nature Communications, however I believe some points should be clarified.

The paper mentions limitations in the calculations leading to differences between computed and experimental EFs. Because of these differences, the increase of 2 orders of magnitude for DFP-4T in experimental EFs is only around 6-21x when using the computed EFs. While differences between calculated and experimental EF can be understood, and even expected, from the arguments presented, the trend between DFPCO-4T and DFH-4T is reversed. Can the authors comment on this discrepancy?

XRD data of DFP-4T give a distance of 4.17Å which does not correspond to any relevant parameter calculated for the crystal. Assuming this distance was representative of the film (not crystal), how would the computed CT quantities and EFs be modified?

Reviewer #2 (Remarks to the Author):

This manuscript reports a SERS active substrate based on a small fluorine substituted organic molecule. The authors attribute the enhancement mechanism to a weakly absorbing charge transfer state involving the molecules and surface. The prospect of an all organic SERS substrate is exciting and the authors attempt to provide a detailed explanation for this effect. They perform spectroscopic characterization of these systems and report enhancement factors of ca. 10^3-5 . This seems promising but I am not sure how unique this response is to the particular system under study. For example, charge transfer interactions between organic and inorganic materials are common which raises the question if this phenomenon is limited to a particular class of materials. All that seems to be required is charge transfer between two organics which is not uncommon. The paper is largely phenomenological in nature although theoretical studies are presented for the sake of

estimating CT energies. I would have liked to have seen a more thorough discussion of the proposed mechanism and the uniqueness of the target materials.

Comments:

- They mention the tendency of DFP-4T molecules to pi stack on the surface as a source of enhancement for Raman cross-sections. This seems reasonable but I wonder if they can elaborate further on the likelihood of a potential plasmonic-type enhancement factor. For example, the delocalized pi-electron densities were cited as a possible avenue for SERS but nothing was discussed mentioning this in the introduction.

- It would be helpful if the authors included absorption spectra in the main text highlighting the proposed charge transfer state. The only evidence was inferred from PL quenching dynamics which can be ambiguous. The combination of two chromophores resulting in quenching is not unique and it is not clear to me how exactly this is related to the proposed SERS mechanism.

- Figure 4a is confusing in terms of the labels. They show a line denoting the excitation wavelength (400 nm) but the respective x-axis intercept shows ~750 nm. Perhaps the labels should be reconfigured to avoid possible confusion?

- Raman enhancement factors are reported but the absolute cross sections at the excitation energy are not provided.

- The authors choose to excite the MB analyte off resonance. Why not try exciting on resonance? Was there too much fluorescence background?

Reviewer #3 (Remarks to the Author):

The authors of this article build on previous exploration of the Raman enhancement properties of thin films of organic semiconductors. The field of surface-enhanced Raman spectroscopy (SERS) is over 40 years old, having originally been associated with Ag or Au nanoparticles for plasmon

resonances coupled to other resonances in the molecule-metal system. The next advance in the field came with the successful exploration of inorganic semiconductors substrates such as metal oxides, sulfides, selenides, nitrides opening a wealth of interesting novel substrates. What these substrates lacked in enhancement power, they made up in stability, cost, and the ability to control more precisely the various resonances in the molecule-substrate system contributing to the enhancement. Only recently have (some of) the current authors pioneered the extension of SERS substrates to organic semiconductors. This area holds considerable promise to extend the range and variability of the possible substrates and the ability to control even more finely the parameters contributing to the enhancement. The authors make considerable use of the theory (both analytical and computational) to guide their choices of substrate.

In this contribution, the authors examine a series of organic semiconductors related to DFP-4T (a fully π -conjugated diperfluorophenyl-substituted quaterthiophene structure) in a thin film. Previous studies on DFH-4T showed enhancement factors around 10³, while the end-capped with π -electron-deficient perfluorophenyl DFP-4T used in this study show an enhancement factor greater than 10⁵. This latter is comparable to many of the early metallic substrate SERS experiments. They attribute the increased enhancement to the (usually weak) charge-transfer transition, which has acquired additional intensity due to vibronic coupling with allowed nearby transitions in the molecule-substrate system. These considerations are augmented by several cogent DFT calculations which reveal the source of the CT transitions and others that pertain to the rather large enhancement. They further show that tuning the location of the CT transition by varying the number of thiophenes (e.g. DFP-nT, where n is varied from 2-5), they can control the location and strength of the CT transition and strongly affect the Raman enhancement.

The authors go a long way towards developing methods to optimize the properties needed for development of sensors using SERS. They are taking advantage of the unique properties of organic semiconductors for Raman enhancement, and are now among the most advanced research groups in this endeavor. This article should definitely be published.

I only have a few comments for improving the work:

Figure 1e "Stacking" is misspelled.

In the DFT calculations, it appears that the substrate is modeled by utilization of only a single molecule. However, one of the important features of the substrate is the collective nature of the organic aggregate. Many such systems have exciton bands, which are formed in the excited state and might have interesting effects on the enhancement properties. It would be of considerable interest if these were explored by appropriate DFT calculations. If one can make DFT calculations in Ag aggregates of over 1000 atoms, certainly good calculations can be made with a number of DFP-4T molecules. We know the crystal structure, and this can be built into the calculations.

Reviewer 1

Reviewer-1 General Comments: This paper demonstrates a notable Raman enhancement factor in films of DFP-4T, where the previously studied DFH-4T has been modified with electron-deficient groups. SERS spectra show the enhancement of the Raman signal. Photoluminescence and quantum-chemical calculations illustrate the role of CT states. Overall I find the data convincing, the significance high in that the EF approaches that of plasmonic systems, and the paper well written. Therefore, I support publication in Nature Communications, however I believe some points should be clarified.

Response: We would like to thank the reviewer for his/her favorable comments on our manuscript supporting the publication in Nature Communications.

Reviewer-1 Comment-1: The paper mentions limitations in the calculations leading to differences between computed and experimental EFs. Because of these differences, the increase of 2 orders of magnitude for DFP-4T in experimental EFs is only around 6-21x when using the computed EFs. While differences between calculated and experimental EF can be understood, and even expected, from the arguments presented, the trend between DFPCO-4T and DFH-4T is reversed. Can the authors comment on this discrepancy?

Response: In addition to the geometric differences between different semiconductor systems mentioned on page 17 that may account for some of the differences in EFs not captured by our computational model, we have added the following sentences to page 18 to address the reviewer's concern about the enhancement factors on **DFPCO-4T** and **DFH-4T** substrates:

“In addition, we compute all enhancement factors on resonance with the CT states; if the CT states in the experimental systems somehow differ in how close to resonance they are with the Raman laser, this will likely cause some deviation in the trends in EFs. We note that our computational model reverses the ordering of the enhancement factors on **DFPCO-4T** and **DFH-4T** relative to their experimental values, likely due to some combination of the semiconductor/**MB** molecular packing parameters and resonance conditions described here.”

Reviewer-1 Comment-2: XRD data of DFP-4T give a distance of 4.17Å which does not correspond to any relevant parameter calculated for the crystal. Assuming this distance was representative of the film (not crystal), how would the computed CT quantities and EFs be modified?

Response: The computational study in this manuscript was performed by electronic structure calculations on semiconductor/**MB** complexes that include a single molecule of the semiconductor and the analyte. Therefore, the intermolecular packing properties of the semiconductor film itself are not taken into account in our computational model. This is mainly because of the high computational cost of the frequency calculations when a large number of semiconductor molecules are included. However, we agree with the reviewer that the packing of the organic substrate is an interesting aspect that definitely deserves further exploration, and this could be the subject of future studies in the field of organic-SERS. To address the reviewer's concern on this matter, the following sentence has been added to page 17:

“Because of the high computational cost of the frequency calculations, computations with more than one substrate molecule are not feasible.”

Reviewer 2

Reviewer-2 General Comments: This manuscript reports a SERS active substrate based on a small fluorine substituted organic molecule. The authors attribute the enhancement mechanism to a weakly absorbing charge transfer state involving the molecules and surface. The prospect of an all organic SERS substrate is exciting and the authors attempt to provide a detailed explanation for this effect.

Response: We would like to thank the reviewer for his/her favorable comments on our manuscript.

Reviewer-2 Comment-1: They perform spectroscopic characterization of these systems and report enhancement factors of ca. 10^3 -5. This seems promising but I am not sure how unique this response is to the particular system under study. For example, charge transfer interactions between organic and inorganic materials are common which raises the question if this phenomenon is limited to a particular class of materials. All that seems to be required is charge transfer between two organics which is not uncommon. The paper is largely phenomenological in nature although theoretical studies are presented for the sake of estimating CT energies. I would have liked to have seen a more thorough discussion of the proposed mechanism and the uniqueness of the target materials.

Response: We would like to thank the reviewer for raising these important points. There are numerous reasons behind the fact that why organic semiconductors are very promising building blocks to design Raman Enhancement platforms: (i) The presence of energetically accessible and delocalized π -electrons in these systems enables unique chemical, physical, and optoelectronic properties as compared to molecules having only σ -electrons. Particularly, solid-state films of properly designed π -conjugated molecules are responsive to electromagnetic radiation in the visible/near-IR spectral region and conduct electrical current under an external electric field, which brings very exciting opportunities for commercialization. (ii) Since the solid state structure of π -conjugated molecules relies on relatively weak intermolecular interactions (e.g., Van der Waals, π - π , and C-H $\cdots\pi$), facile film fabrication to form various micro/nano-structures is feasible by using simple solution- or physical vapor deposition (PVD)-based

techniques. (iii) π -Conjugated systems could be biocompatible and they could have very low materials production costs once their synthesis processes are optimized. (iv) Highly delocalized molecular orbitals of π -conjugated systems ensure efficient orbital overlap with the analyte molecules, which could facilitate intermolecular charge transfer processes with the analyte molecules to contribute to the chemical enhancement mechanism in SERS. (v) The possibility to tune organic semiconductor carrier density by molecular and field-effect doping processes, thus opening new possibilities for new sensor device architectures. (vi) Very different than universal coinage metal surfaces, organic semiconductor structures could also be specifically designed to detect a particular analyte molecule, which could lead to molecular-specific SERS platforms. To achieve large SERS enhancements, the molecular structure of organic semiconductors should be carefully designed and must provide finely tuned π -molecular orbital properties, electronic structures, charge transfer properties, and solid-state packing motifs, which are expected to affect the SERS activity of the organic semiconductor. Considering all these important points, very recently we demonstrated that metal-free nanostructured semiconductor films of a fluorocarbon-substituted α,ω -diperfluorohexylquaterthiophene, **DFH-4T**, could be SERS-active with a SERS enhancement factor surpassing 10^3 . (*M. Yilmaz, E. Babur, M. Ozdemir, R. L. Giesecking, Y. Dede, U. Tamer, G. C. Schatz, A. Facchetti, H. Usta, G. Demirel, Nat. Mater. 2017, 16, 918-924.*) Prompted by this breakthrough, in this work, we envisioned to design a fluoroarene-modified oligothiophene, 5,5''-diperfluorophenyl-2,2':5',2'':5''':2''''-quaterthiophene (**DFP-4T**) semiconductor, which includes an electron-rich quaterthiophene π -core similar to that of **DFH-4T** but end-capped with π -electron-deficient perfluorophenyl ($-C_6F_5$) units. Thus, this molecule employs a quite different design approach as compared with **DFH-4T**, wherein the σ -insulating perfluoroalkyl substituents are replaced with $-C_6F_5$ substituents leading to a fully π -conjugated backbone with small intramolecular torsions. Furthermore, this molecular design enhances solid-state molecular packing and correspondingly favors very efficient charge-transfer/transport in films and facilitate frontier orbital wave function overlap with the analyte molecules to further enhance SERS performance. We kept fluorine substituents on the outer benzene rings to continue to provide excellent volatility for reliable and quantitative PVD film fabrication. As a result, nanostructured **DFP-4T** films prepared via PVD are found to exhibit unprecedented EF values of $>10^5$, indicating that the SERS performance of metal-free organic semiconductor films could approach those of the current plasmonic metal and inorganic semiconducting platforms.

At this point, we believe that π -conjugated frameworks could be designed to yield either universal or molecule-specific SERS platforms, which would totally depend on the specific chemical structures of the semiconductor and analyte molecules. This point was also discussed by J. R. Lombardi and it's mentioned as one of the key advantages of organic semiconductor SERS platforms as compared with universal coinage metal surfaces (*Nature Materials*, 2017, 16, 878-880). However, considering that the organic SERS platform is a very new research area and there are only two examples of SERS-active organic π -conjugated semiconductors, which are both reported by us, it is not straightforward at this point to address the reviewer's question regarding the uniqueness. That would require a separate detailed study on a number of analyte samples with varied chemical structures, which is beyond the scope of the current manuscript.

The Reviewer also would like to see more detailed discussion on the enhancement mechanism of the organic semiconductor platforms. Considering that our organic semiconductor films have a low intrinsic carrier density (*G. Demirel, H. Usta, M. Yilmaz, M. Celik, H. A. Alidagi, F. Buyukserin, J. Mater. Chem. C*, 2018, 6, 5314-5335) and negligible plasmon resonances in the conduction band, the significant Raman enhancements observed for **DFP-4T** films are very unlikely to originate from the electromagnetic mechanism. Additionally, both our analyte (**MB**) and organic semiconductor films do not exhibit any significant optical transition at the laser excitation energy, neither in solution nor solid-state. Thus, the observed impressive enhancement factors of our platforms cannot be simply explained by resonant charge excitation processes on pristine analyte molecular aggregates or the nanostructured organic films. However, alternative energetically feasible processes may occur for hybridized analyte/organic semiconductor electronic states at the analyte/semiconductor interface, which eventually contributes to the magnification of the Raman scattering cross-sections. To gain insight into the Raman enhancement mechanism in these systems, we have computed the excited-state properties of complexes formed by the analyte (**MB**) and organic semiconductors at the INDO/SCI (intermediate neglect of differential overlap/single configuration interaction) level, which has rationalized the SERS enhancement mechanism on plasmonic metals. (*R. L. Giesecking, M. A. Ratner, G. C. Schatz, Faraday Discuss.* 2017, 205, 149-171) For all complexes, the first excited state has an energy of 1.35 eV for **MB/DFP-4T**, 1.62 eV for **MB/DFPCO-4T**, and 1.43 eV for **MB/DFH-4T**, respectively. In all three complexes, the CT excited states have >95% charge-transfer character from the HOMO of the semiconductor molecule to the LUMO of **MB**.

Interestingly, the CT excited state energies strongly correlate with the substrate HOMO energies. Within computational error, these are close to the experimental SERS excitation energy (1.58 eV), given the simplifications of the INDO model, suggesting that resonance CT enhancement of the Raman signal is a likely source of the experimentally observed SERS. Additionally, the high sensitivity of the CT state to molecular geometry allows this state to contribute strongly to the resonance differential polarizability even though its absorption is weak. Since the oscillator strength strongly depends on the substrate–analyte wavefunction overlap, these results also suggest that obtaining large enhancements is possible only for substrate morphologies that allow significant π -stacking of the analyte with the substrate molecules. The key role of CT state absorption in the Raman enhancement is in line with our experimental observation that a 3D nanostructured organic semiconductor morphology leads to an effective chemisorption of the analyte molecules with proper molecular orientations to facilitate CT. We also note that the porous/shaped 3D morphology of organic semiconductor films would also provide electromagnetic enhancements (probably by a factor of ~ 10) (S. Zhou, G. C. Schatz, *Chem. Phys. Lett.* 403, 62-67, 2005); however, this cannot be the dominant effect as it would lead to much larger intensities for **DFP-4T** than for **MB**, which is not what is observed in Figure 2.

As suggested by the reviewer, we have now extended the related discussion about the enhancement mechanism (pages 12-19) and the potential uniqueness of the target molecules (pages 3, 4 and 19) in the revised manuscript.

Reviewer-2 Comment-2: They mention the tendency of DFP-4T molecules to pi stack on the surface as a source of enhancement for Raman cross-sections. This seems reasonable but I wonder if they can elaborate further on the likelihood of a potential plasmonic-type enhancement factor. For example, the delocalized pi-electron densities were cited as a possible avenue for SERS but nothing was discussed mentioning this in the introduction.

Response: We would like to thank the reviewer for raising this important point. As mentioned in the manuscript, the molecule employed in this work (**DFP-4T**) is an intrinsic organic semiconductor with a very low intrinsic carrier density of $<10^{15} \text{ cm}^{-3}$ and negligible plasmon resonances in the conduction band. Therefore, when pristine **DFP-4T** films are implemented as SERS platforms, despite their unique morphology defined by a rough and nanostructured

surface, the electromagnetic mechanism could not play any role. For the current and previous semiconductors studied by us as SERS platforms, chemical enhancement relying on charge-transfer processes between the analyte and semiconductor molecules remains to be the major SERS mechanism. However, it is known in the literature that localized surface plasmons could be generated in non-metallic structures such as inorganic semiconductors (*J. Phys. Chem. Lett.*, 2014, 5, 976-985), and some of these strategies could also be applied to organic semiconductors. To this end, chemical doping or electrical charge-injection (under gate-bias for example) could tailor band edges and introduce free charge carriers into organic semiconductor structures, which could lead to the formation of localized surface plasmons. This could be potentially useful to further advance the SERS enhancement factors of organic semiconducting platforms. As suggested by the reviewer, we have now discussed these possibilities and potentials in the discussion part of the revised manuscript to provide some guidelines for future studies on organic-SERS platforms (page 19).

Reviewer-2 Comment-3: It would be helpful if the authors included absorption spectra in the main text highlighting the proposed charge transfer state. The only evidence was inferred from PL quenching dynamics which can be ambiguous. The combination of two chromophores resulting in quenching is not unique and it is not clear to me how exactly this is related to the proposed SERS mechanism.

Response: The optical absorption spectra for **MB** in aqueous solution/solid state and pristine/MB-deposited **DFP-4T** films are shown in Supplementary Figure 4, which do not show any strong optical excitation to a CT state near the Raman excitation wavelength of 785 nm. Our computational results at the INDO/SCI level suggest that the SERS enhancements in **MB/DFP-4T** and **MB/DFPCO-4T** are due to the presence of charge-transfer (CT) excited states with a small but non-zero absorption cross-section near resonance with the Raman laser. Therefore, this CT state absorbs light very weakly, which does not allow us to observe it in the optical absorption spectra. However, this weak absorption is sufficient to cause large resonance Raman enhancement because the large change in charge density upon excitation leads to large changes in resonant polarizability derivatives with respect to vibrational motion. These observations are consistent with our previous system of **MB** on **DFH-4T**. (*Nature Materials*, 2017, 16, 918–924) Therefore, we believe that the optical absorption spectra shown in Supplementary Figure 4

provides only supplementary information and, with all due respect to the reviewer's suggestion, we would like to keep it in the Supporting Information. On another note pointed out by the reviewer regarding the PL study, the transient PL spectra for both samples were measured in different positions and **MB** deposition was found to consistently reduce the PL lifetimes at short delays (please see Figure 4b and 4c). This suggests that although emissive **MB** states are not populated, the presence of **MB** on organic semiconductor surfaces provides additional decay pathway(s) for the photoexcitation, presumably via charge-transfer involving **MB** electronic states. Energy transfer can be excluded as there is no signature of the **MB** photoluminescence in the spectra. Therefore, these steady-state and transient PL studies provide us with additional information, which could not be obtained from optical absorption spectra, for the formation of charge-transfer states. These CT states, as supported by the computational results, are involved in the observed SERS mechanism, and this is discussed in detail in the manuscript (pages 12-19).

Reviewer-2 Comment-4: Figure 4a is confusing in terms of the labels. They show a line denoting the excitation wavelength (400 nm) but the respective x-axis intercept shows ~750 nm. Perhaps the labels should be reconfigured to avoid possible confusion?

Response: We would like to thank the reviewer for his/her careful reading of the manuscript and raising this point. The steady state photoluminescence spectrum in Figure 4a shows three shaded areas centered at 580 nm, 620 nm and 750 nm corresponding to the wavelengths at which photoluminescence decay profiles are detected. These decay profiles are shown in Figures 4b-4d, and during these measurements the excitation wavelength is at 400 nm. Therefore, the label of "400 nm excitation" does not correspond to any shaded area. However, as pointed out by the reviewer, we have now removed the label "400 nm excitation" from Figure 4a to avoid possible confusion, and we have revised the caption of Figure 4 to clarify the meaning of these shaded areas.

Reviewer-2 Comment-5: The authors choose to excite the MB analyte off resonance. Why not try exciting on resonance? Was there too much fluorescence background?

Response: As the reviewer points out, the SERS measurements performed on the nanostructured **DFP-4T** film for the **MB** probe molecule are non-resonant and it gives strong Raman signals without any fluorescence background. This is consistent with the fact that drop-casted **MB** on the nanostructured **DFP-4T** film does not show any optical absorption at the Raman excitation

wavelength of 785 nm to yield electronic excitation (please see Supplementary Figure 4). Since the utilization of organic semiconductors in the field of SERS is a new research area and there are still a number of facts that need to be clarified about their enhancement mechanisms, a non-resonant probe molecule is preferred in our study. In this way, (as stated on page 8 of the manuscript) the contribution of the probe molecule's pure resonant excitation to the Raman enhancement could be excluded and the role of the nanostructured organic semiconductor surface on the SERS mechanism could be better understood. If a Raman reporter having an optical transition at the Raman laser excitation wavelength was employed in the experiments, we could not make sure whether the observed Raman enhancements originate from the probe molecule's electronic excitation or it relies on the intermolecular interactions between the probe and the organic semiconductor. However, note that we are still in the process of studying and understanding key structural and morphological features of organic semiconductor surfaces that enable the unprecedented SERS effects. When these points are further clarified in future studies, the applicability of organic semiconductor based platforms in SERS could be extended to various analyte molecules to include both resonant and non-resonant SERS measurements.

Reviewer-2 Comment-6: Raman enhancement factors are reported but the absolute cross sections at the excitation energy are not provided.

Response: We thank the reviewer for his/her comment. In order to calculate the SERS enhancement factor, it is not necessary to know the absolute cross-section of the analyte. Instead, measuring relative Raman intensities and number of molecules in the laser spot areas is sufficient as this is a typically used method in the SERS literature. However, to address the reviewer's concern regarding the absolute cross section of **MB** on nanostructured **DFP-4T** surface, we have performed an additional experiment using Rhodamin 6G (**R6G**) and **MB** analyte molecules using the same experimental set-up. **R6G** was selected as a standard to determine the cross-section of **MB**, and its Raman peak at 1510 cm^{-1} was used as a reference with a non-SERS absolute cross-section of $9.6 \times 10^{-27}\text{ cm}^2/\text{sr}$ (E. C. Le Ru, E. Blackie, M. Meyer, P. G. Etchegoin, *J. Phys. Chem. C* 2007, 111, 137-13803) at 785 nm (extrapolated from 633 nm, assuming absolute cross-section is proportional to $\nu_s^3 \nu_L$) (M. J. Colles, J. E. Griffiths, *J. Chem. Phys.* 1972, 56, 3384-3391). The absolute SERS cross-section of **MB** was then determined as $2.4 \times 10^{-24}\text{ cm}^2/\text{sr}$ for the 1621 cm^{-1}

Raman peak. As suggested by the reviewer, we have now included the absolute cross section calculation details in the revised manuscript (page 21).

Reviewer 3

Reviewer-3 General Comments: The authors of this article build on previous exploration of the Raman enhancement properties of thin films of organic semiconductors. The field of surface-enhanced Raman spectroscopy (SERS) is over 40 years old, having originally been associated with Ag or Au nanoparticles for plasmon resonances coupled to other resonances in the molecule-metal system. The next advance in the field came with the successful exploration of inorganic semiconductor substrates such as metal oxides, sulfides, selenides, nitrides opening a wealth of interesting novel substrates. What these substrates lacked in enhancement power, they made up in stability, cost, and the ability to control more precisely the various resonances in the molecule-substrate system contributing to the enhancement. Only recently have (some of) the current authors pioneered the extension of SERS substrates to organic semiconductors. This area holds considerable promise to extend the range and variability of the possible substrates and the ability to control even more finely the parameters contributing to the enhancement. The authors make considerable use of the theory (both analytical and computational) to guide their choices of substrate. The authors go a long way towards developing methods to optimize the properties needed for development of sensors using SERS. They are taking advantage of the unique properties of organic semiconductors for Raman enhancement, and are now among the most advanced research groups in this endeavor. This article should definitely be published.

Response: We would like to deeply thank the reviewer for his/her favorable comments and support for publication in Nature Communications.

Reviewer-3 Comment-1: Figure 1e “Stacking” is misspelled.

Response: We would like to thank the reviewer for his/her careful reading of the manuscript. The typo in Figure 1e has now been corrected as suggested by the reviewer.

Reviewer-3 Comment-2: In the DFT calculations, it appears that the substrate is modeled by utilization of only a single molecule. However, one of the important features of the substrate is the collective nature of the organic aggregate. Many such systems have exciton bands, which are formed in the excited state and might have interesting effects on the enhancement properties. It would be of considerable interest if these were explored by appropriate DFT calculations. If one can make DFT calculations in Ag aggregates of over 1000 atoms, certainly good calculations can be made with a number of DFP-4T molecules. We know the crystal structure, and this can be built into the calculations.

Response: We agree with the reviewer that the packing of the organic substrate is an interesting aspect that definitely deserves further exploration. However, the reviewer's suggestion of performing calculations on **DFP-4T** aggregates is not feasible for our purposes. Although ground-state DFT calculations on hundreds of atoms are possible, particularly in highly symmetric systems or when other simplifications are made, the frequency-dependent Raman intensities are much more computationally intensive and are not feasible for systems of that size. To address these concerns, the following sentence has been added to page 17: "Because of the high computational cost of the frequency calculations, computations with more than one substrate molecule are not feasible."

REVIEWERS' COMMENTS:

Reviewer #1 (Remarks to the Author):

The authors have clarified the possible reasons for the discrepancies between experiments and simulations. In their response to other reviewers, the authors also discussed the role of CT states and other potential mechanisms in the SERS enhancement and comment on the uniqueness of the observed effects.

I therefore favor publication.

Reviewer #2 (Remarks to the Author):

The authors have done a thorough job of addressing my previous concerns with the manuscript. They try to offer a more detailed explanation for the proposed SERS type Raman scattering enhancement. I still feel some aspects are very preliminary but it will be interesting and insightful to further examine this effect. Overall, I feel the paper is now publishable.

Reviewer #3 (Remarks to the Author):

I have reviewed the changes made by the authors in response to the comments of the referees, and I believe the changes made by the authors are responsive and adequate. I believe this article should be published.